# Exploring the Link Between PACAP Signalling and Hyaluronic Acid Production in Melanoma Progression

**DOI:** 10.3390/ijms262412049

**Published:** 2025-12-15

**Authors:** Tibor Hajdú, Patrik Kovács, Éva Katona, Minh Ngoc Nguyen, Judit Vágó, Csaba Fillér, Róza Zákány, Gabriella Emri, Gábor Tóth, Dóra Reglődi, Tamás Juhász

**Affiliations:** 1Department of Anatomy, Histology and Embryology, Faculty of Medicine, University of Debrecen, Nagyerdei krt. 98, H-4032 Debrecen, Hungary; hajdu.tibor@med.unideb.hu (T.H.); patrik.kovacs@med.unideb.hu (P.K.); katona.eva@anat.med.unideb.hu (É.K.); minh.ngoc.nguyen@med.unideb.hu (M.N.N.); vago.judit@med.unideb.hu (J.V.); filler.csaba@med.unideb.hu (C.F.); roza@anat.med.unideb.hu (R.Z.); 2Department of Dermatology, Faculty of Medicine, University of Debrecen, Nagyerdei krt. 98, H-4032 Debrecen, Hungary; gemri@med.unideb.hu; 3Department of Medical Chemistry, Albert Szent-Györgyi Medical School, University of Szeged, Dom tér 8, H-6720 Szeged, Hungary; toth.gabor@med.u-szeged.hu; 4Department of Anatomy, HUN-REN-PTE PACAP Research Team, Medical School, University of Pécs, Szigeti út 12, H-7624 Pécs, Hungary; dora.reglodi@aok.pte.hu

**Keywords:** PACAP, SOX9, SOX10, hyaluronan, HAS, RHAMM, MITF, Hyal

## Abstract

Pituitary adenylate cyclase-activating polypeptide (PACAP) is a small neuropeptide detected first in the hypothalamo–hypophyseal system; recently, it has also been identified in peripheral organs and in tumours. It is well demonstrated that PACAP exerts cell- and tissue-protecting effects in various stressful conditions and helps to maintain tissue homeostasis. In melanoma, the anti-invasive effect of PACAP has been demonstrated; however, there is also existing sporadic data which proves PACAP plays a role in melanoma progression. The major goal of our study was to investigate the signalling targets of PACAP in A2058 and WM35 melanoma cell lines in vitro. Various molecular players of melanocyte differentiation and function responded to PACAP application. SOX9 expression increased while SOX10 expression decreased and CREB signalling did not change. The expression level of TYRP1 decreased, while DCT elevated, and MITF expression showed changes at the mRNA level and in its subcellular localisation. In contrast, the amount of hyaluronan (HA) and expressions of its synthases, as well as RHAMM, increased, indicating the role of PACAP in secretion of an HA-rich matrix. In parallel with these results, we detected elevated hyaluronidase2 (Hyal2) expression in the presence of PACAP. On the other hand, alfaV and beta3 integrin expressions did not alter significantly. Our results demonstrate that exogenous PACAP modulates the expression of multiple target molecules in melanoma cells. Some of the significantly responding molecules take part in hyaluronan homeostasis, suggesting an effect of PACAP on tumour matrix composition, through which it can modulate invasiveness of melanoma cells.

## 1. Introduction

Melanoma of the skin is a malignant tumour arising from epidermal melanocytes. Although cutaneous melanoma accounts for only a minor proportion of all skin tumours, it is responsible for the vast majority of deaths caused by skin-related malignant tumours [1,2]. Melanocytes originate from the neural crest and migrate into the epidermis during embryonic life. Motility of mature melanocytes becomes restricted, and together with the proliferation ability of the pigment cells, is under the control of epidermal keratinocytes. Nonetheless, the high invasiveness and strong metastasis-forming property of malignantly transformed pigment cells stem from the intense motility of the undifferentiated embryonic melanocyte precursors [3]. Melanocytes differentiate early in development from melanoblasts arising from neural crest cells—a transient, highly migratory, stem-like population—and these melanoblasts reach their destinations in the skin, eye, and other tissues by using epithelial–mesenchymal–transition (EMT)-like programmes and dynamic cytoskeletal remodelling, exhibiting high motility and invasiveness through extracellular matrices and chemotactic cues such as endothelin, Wnt, and KIT ligands. Melanoma cells exploit and reactivate these same neural crest-derived migratory programmes by switching from a proliferative MITF-high state to an invasive MITF-low state, and this transition drives migration through the extracellular matrix, intravasation/extravasation, and ultimately early metastatic spread [4]. In Europe, the overall epidemiological trends highlight the alarming increase in cutaneous melanoma incidence and mortality [2,5,6,7]. Early detection and advancements in treatment have improved the prognosis, yet melanoma remains a formidable challenge due to its aggressive nature and propensity for metastasis [2,8].

The primary risk factor for melanoma is exposure to ultraviolet (UV) radiation. UV radiation causes skin inflammation and may cause DNA damage, leading to mutations that can result in melanoma [9]. Genetic predisposition plays a pivotal role in melanoma risk [6]. One important interaction is that UV radiation can have a stronger carcinogenic effect in individuals with inherited susceptibility—such as pathogenic variants in CDKN2A, MC1R, or other genes involved in pigmentation and DNA repair—which increase sensitivity to UV-induced DNA damage and reduce the efficiency of its repair, thereby amplifying melanoma risk beyond the effect of UV exposure alone [10]. The tumour microenvironment, consisting of various cell types and extracellular matrix (ECM) components, plays a crucial role in melanoma progression and metastasis [11]. Signalling molecules present in the tumour microenvironment can profoundly influence tumour behaviour. Moreover, melanoma cells can manipulate their microenvironment to promote angiogenesis and evade immune detection [12].

Melanoma is known for its aggressive nature and propensity to metastasise to distant organs, even from a relatively small primary lesion. Key factors that enhance the metastatic potential of melanoma include the expression of adhesion molecules, proteolytic enzymes, and the ability to adapt to different tissue environments [13]. Signalling molecules within the melanoma tumour microenvironment, such as those involving the MAPK, PI3K/AKT, and Wnt pathways, can profoundly influence tumour behaviour and melanoma cells actively manipulate these pathways to remodel their microenvironment, promoting angiogenesis, sustaining growth, and evading immune detection [14]. Understanding these signalling pathways is essential for identifying potential therapeutic targets and improving treatment outcomes [14].

Recent studies have highlighted the importance of the extracellular matrix (ECM) in shaping the tumour microenvironment during melanomagenesis [15]. Hyaluronic acid (HA), a major ECM component composed of repeating units of glucuronic acid and N-acetylglucosamine, has gained particular attention for its direct influence on melanoma cell behaviour. HA exists in multiple molecular weights, with high-molecular-weight (HMW) HA generally exerting anti-inflammatory and tissue-protective effects, whereas low-molecular-weight (LMW) HA—often generated during tumour-associated tissue remodelling—can promote melanoma cell migration, proliferation, angiogenesis, and EMT-like changes that drive invasion and metastatic spread [16,17]. Through receptors such as CD44 and RHAMM, HA activates signalling pathways including MAPK/ERK that support melanoma cell survival and motility, and elevated HA–CD44 signalling is associated with more aggressive melanoma phenotypes [17,18,19,20,21,22]. Hyaluronidases (HYAL1–4), the enzymes responsible for degrading HA, contribute to melanoma progression by generating low-molecular-weight HA fragments that enhance melanoma cell motility, invasion, and metastatic potential, and by modulating CD44- and RHAMM-mediated signalling pathways that support a pro-tumourigenic microenvironment [23].

Pituitary adenylate cyclase-activating polypeptide (PACAP) is a 38-amino-acid neuropeptide structurally related to VIP and best known for roles in neuroprotection, neurodevelopment, and immune modulation [24]. PACAP signals primarily through the high-affinity PAC1 receptor and to a lesser extent, through VPAC1 and VPAC2 [25], activating intracellular pathways such as adenylate cyclase/cAMP–PKA, PLC–IP_3_/DAG–PKC, and multiple MAPK cascades (ERK, JNK, p38) [24,26,27]. These same pathways are critically involved in melanoma cell survival, proliferation, plasticity, and motility. Emerging evidence suggests that PACAP–PAC1 signalling may influence melanoma progression by enhancing MAPK- and cAMP-driven transcriptional programmes that support tumour growth, by modulating cytoskeletal dynamics through PKC and calcium signalling and by interacting with immune-regulatory mechanisms that can facilitate tumour invasion and immune evasion. Consequently, dysregulated PACAP signalling within the melanoma microenvironment may contribute to increased aggressiveness and metastatic behaviour.

PACAP signalling has been implicated in cancer biology. It affects cell proliferation and survival and can contribute to tumour growth and metastasis [28]. Recently, we have shown that PACAP and its receptors are expressed in various melanoma cell lines and tumour samples [29]. The expression levels of PACAP and its receptors can correlate with melanoma progression and patient prognosis, suggesting a role in tumour development and maintenance [29]. Moreover, PACAP’s role in modulating the immune response may influence tumour immune evasion [30]. This immunomodulatory role is critical in the context of melanoma, where the immune system’s ability to recognise and attack tumour cells is often compromised [31]. On the other hand, SOX9, SOX10, and MITF are key transcription factors in melanocyte development and function. Although no published work states that PACAP can regulate these transcription factors in melanoma progression, their signalling connection may play important role in melanoma formation. SOX10 promotes neural crest-derived melanocyte lineage commitment and drives the expression of melanocyte-specific genes, while SOX9 contributes to melanocyte differentiation and stress responses [32]. MITF acts as a master regulator of melanocyte biology, controlling pigmentation, survival, proliferation, and cellular plasticity and its activity integrates signals from multiple developmental and oncogenic pathways [33].

It is known that an HA-rich ECM influences the motility of cells [34]. We have shown that PACAP increased ECM production, particularly HA synthesis, during chondrogenesis [35]. We also published data indicating that PACAP treatment significantly reduced spontaneous motility of NHEM and melanoma cells; moreover, chemotaxis and Matrigel invasion of melanoma cells also were strongly reduced after PACAP application [29]. Nonetheless, further research is needed to elucidate the potential interplay between PACAP and HA in the context of melanoma development and progression. In the present work, we aimed to investigate the influence of PACAP on HA homeostasis and melanocyte differentiation factors’ expression of melanocytes and melanoma cells. The data presented in this study underline the significance of PACAP signalling in HA homeostasis of melanoma cells, suggesting that it may be one of the factors causing the restriction of melanoma cell motility. We also demonstrate significant alterations in the expression levels of SOX9, SOX10, and MITF transcription factors, which are crucial elements in melanocyte differentiation.

## 2. Results

### 2.1. Altered PACAP Receptor Expression in the Presence of Neuropeptides

PACAP 1-38 was administered to the medium of melanocytes, A2058, and WM35 melanoma cells. The expression of PACAP receptors PAC1-R and VPAC1-R has previously been demonstrated on human epidermal melanocytes, as well as on A2058 and WM35 melanoma cells [29]. In this study, the administration of PACAP 1-38 altered the mRNA expression of these receptors. In melanocytes and both melanoma cell lines, PAC1-R mRNA showed a slight, but not significant, elevation in the presence of the neuropeptide, while the mRNA expression of VPAC1-R was reduced (Figure 1A).

At the protein level, the expression of PAC1-R was elevated in melanocytes and A2058 cells, with only a slight increase detected in WM35 cells (Figure 1B). Furthermore, PACAP 1-38 augmented the protein expression of VPAC1-R in melanocytes and A2058 cells but reduced it in the WM35 cell line (Figure 1B). The mRNA expression of DPP4, the primary degrading enzyme of PACAP, was reduced in melanocytes but elevated in both melanoma cell lines (Figure 1A). In contrast, its protein expression showed a significant elevation in all cell lines after PACAP treatment (Figure 1B). Immunocytochemistry revealed that DPP4 cytoplasmic localisation increased after PACAP 1-38 administration in every cell line examined (Figure 1C).

### 2.2. Possible Downstream Target of PACAP Signalling in Melanoma Cells

As the canonical downstream target of the PAC1-R is adenylate cyclase, which induces the activation of PKA, the expression of PKA and its potential target transcription factors (CREB, SOX9, SOX10, and MITF) was investigated. Although PACAP administration did not significantly alter PKA mRNA expression in either melanoma cell line or melanocytes (Figure 2A), the total protein expression of PKA increased in both melanoma cell lines, while no change was detected in epidermal melanocytes (Figure 2B). The active, phosphorylated form of the enzyme (p-PKA) was elevated in melanoma cells and melanocytes as well, after PACAP administration, indicating the increased activity of the canonical PACAP signalling pathway (Figure 2B).

Next, we investigated the influence of PACAP on the main signal molecules regulating melanocyte differentiation. Following PACAP treatment, the mRNA expression of the transcription factor CREB remained constant in both melanoma cell lines but was reduced in melanocytes (Figure 2A). In contrast, CREB protein expression showed a significant increase in melanoma cell lines, and a slight decrease was detected in melanocytes after PACAP administration (Figure 2B). Interestingly, the level of its more active phosphorylated form (p-CREB) did not change in the presence of the neuropeptide in either cell line (Figure 2B).

The mRNA expression of *Sox9* did not change significantly in melanoma cell lines but decreased in melanocytes (Figure 2A). In contrast to the mRNA levels, SOX9 protein expression showed a significant increase in the WM35 cell line (Figure 2B), while its active phosphorylated form (p-SOX9) showed a significant elevation in all cell types after PACAP administration (Figure 2B). For *Sox10*, PACAP administration decreased its mRNA expression in the A2058 cells and melanocytes, with no alteration in WM35 cells (Figure 2A). At the protein level, SOX10 was significantly reduced in both melanoma cell lines and in melanocytes (Figure 2B).

Although the mRNA expression of *MITF* was elevated in both melanoma cell lines following PACAP treatment, a significant reduction was detected in melanocytes (Figure 2A). At the protein level, MITF did not significantly change in either melanoma cell line but was reduced in melanocytes (Figure 2B). The subcellular localisation of MITF was also observed. In control cells, MITF was predominantly localised to the cytoplasm with barely detectable nuclear staining (Figure 2C). PACAP treatment increased the nuclear translocation of MITF in both melanoma cell lines (Figure 2C). In melanocytes, weak signals were detected either in the cytoplasm or nucleus, which were not significantly altered by PACAP (Figure 2C).

Finally, the expression of the melanogenesis-related enzymes DCT and TYRP1 were examined. No alterations were demonstrated in the mRNA expression of *DCT* in any cell line (Figure 2A). In contrast, while DCT protein expression was barely detectable in control cells, it significantly increased in both malignant cell populations in the presence of PACAP, whereas in melanocytes, DCT expression was significantly decreased (Figure 2B). The mRNA expression of TYRP1 showed a slight, non-significant reduction in all cell lines (Figure 2A), while its protein expression decreased significantly after PACAP treatment in all cell cultures (Figure 2B).

### 2.3. Effects of PACAP on HA Production and Its Receptors

Using affinity cytochemistry, HA was detectable in the cytoplasm of both WM35 and A2058 cells, as well as in pericellular and intercellular locations; moreover, intranuclear signal was also demonstrated (Figure 3A). In epidermal melanocytes, pericellular and cytoplasmic staining was detectable, but nuclear signals were absent (Figure 3A). Following PACAP treatment, an increase in intercellular HA was observed (Figure 3A). Fluorescence intensity analysis confirmed that PACAP increased HA levels in both A2058 and WM35 melanoma cells and in melanocytes (Figure 3B).

The protein expression of the HA receptor RHAMM significantly increased in A2058 and WM35 cells in the presence of PACAP, with only a slight elevation detected in melanocytes (Figure 3C). In contrast, the protein expression of CD44 was not significantly altered in either melanoma cell line and similarly to it no change was detected in melanocytes (Figure 3C). By immunocytochemistry, RHAMM immunopositivity was primarily observed intracellularly, surrounding the nuclei of melanoma cells (Figure 3D). In epidermal melanocytes, RHAMM expression was near the detection limit (Figure 3D). In the presence of PACAP, an increase in the cytoplasmic and membrane-associated localisation of RHAMM was observed in melanoma cells, while in melanocyte cultures, some strongly RHAMM-positive cells appeared (Figure 3D). Weaker CD44 signals were detected in the cytoplasm of all cells, with a slight increase in intracellular localisation in melanoma cells following PACAP treatment (Figure 3E).

### 2.4. PACAP Altered the Expression of Enzymes Involved in Hyaluronan Metabolism

The expression of hyaluronan synthases (HASs) was investigated. The protein expression of HAS1 was not detectable in the melanoma cell lines but a strong expression was visible in epidermal melanocytes, where PACAP treatment reduced its expression (Figure 4A). The protein expression of HAS2 and HAS3 was minimal in either control melanoma cells or melanocytes, but it dramatically increased in the presence of PACAP in all cases (Figure 4A). Immunocytochemistry revealed a strong intracellular elevation in HAS2 expression after PACAP treatment, and some nuclear signals also were observed (Figure 4B). For HAS3, strong intracellular and nuclear signals were seen in all cell lines, and PACAP administration strongly increased this immunopositivity in both compartments (Figure 4C).

The expression of HA-degrading hyaluronidases (HYALs) was also examined. The expression of HYAL1 was not detectable in any cell line (Figure 4D). A high expression of HYAL2 was demonstrated in malignant cells, which was further elevated by PACAP, while no expression was found in melanocytes (Figure 4D). Inversely, HYAL3 was not detectable in melanoma cells but was strongly expressed in melanocytes, where its expression was further elevated by PACAP (Figure 4D). The expression of HYAL4 was near the detection limit in all cells, with a low but significant elevation observed in the presence of PACAP (Figure 4D).

### 2.5. Integrin Receptor Expression in Melanoma Cells

To investigate potential effects on cell adhesion, the expression of integrin αV and integrin β3 was monitored. The mRNA expression did not change significantly in the A2058 and WM35 cell lines (Figure 5A). In epidermal melanocytes, however, *integrin αV* mRNA expression significantly decreased after PACAP addition, while no alteration was detected in *integrin β3* expression (Figure 5A). At the protein level, the expression of these integrins also did not significantly change in the presence of PACAP in either melanoma cell line (Figure 5B). In contrast, in melanocytes, integrin αV protein expression significantly decreased after PACAP addition, while a significant elevation in integrin β3 protein was detected (Figure 5B).

## 3. Discussion

Originally, pituitary adenylate cyclase-activating polypeptide (PACAP) was described as an important factor with central role in neuroprotection, regulation of neuroendocrine secretion, and modulation of immune responses.

Recently, PACAP gained attention in the context of cancer biology, particularly regarding tumour progression. PACAP exerts its effects interacting with specific receptors, PAC1, VPAC1, and VPAC2, which are demonstrated in melanoma cells both in vivo and in vitro [29]. Upon binding to these receptors, PACAP activates intracellular signalling cascades, primarily through the cAMP-production and protein kinase pathways [24]. As melanocytes are of neural crest origin, malignantly transformed pigment cells may exhibit altered PACAP signalling. Because previous studies have reported contradictory effects of PACAP on tumour cell behaviour, our aim in this work was to characterise PACAP’s influence on its downstream signalling pathways and on melanocyte differentiation markers.

In our experiments, exogenous PACAP robustly enhanced hyaluronan (HA) synthesis, as demonstrated by the increased HA signal detected through bHABC staining and quantitative image analysis. This HA elevation indicates that PACAP can modulate the tumour microenvironment by stimulating HA production in melanoma cells, which is consistent with reports suggesting that PACAP promotes tumour progression via multiple mechanisms [36]. PACAP has been shown to influence the tumour microenvironment by promoting angiogenesis, thereby enhancing the supply of nutrients and oxygen to tumour cells [37]. PACAP was able to influence the behaviour of surrounding stromal [38] and immune cells [39], potentially creating an environment that supports melanoma cell migration and invasion. However, in contrast to the pro-invasive implications of increased HA and prior literature, our previous functional assays showed that PACAP actually reduced the migratory and invasive capabilities of A2058 and WM35 melanoma cells [29]. Thus, in our model, PACAP-driven HA upregulation does not translate into a pro-invasive phenotype. Instead, PACAP appears to exert overall anti-invasive effects, despite its ability to elevate HA levels. This apparent discrepancy suggests that PACAP’s impact on melanoma invasiveness is context-dependent, and that the increase in HA observed in our study may reflect microenvironmental modulation rather than a direct enhancement of invasive behaviour.

In our current experiments, the expression of the most potent receptor PAC1 is increased while the VPAC1 receptor expression did not alter upon addition of PACAP (Figure 6). Moreover, the expression of DDP4, a PACAP degrading enzyme, increased as well. Others also found that PACAP exerted a positive effect on its receptor expression [40]. This upregulation of the PAC1 receptor by PACAP could, in principle, influence downstream signalling events, including activation of AC and PKA. However, although PAC1 upregulation may increase PKA phosphorylation, our current data do not provide direct evidence that PKA activation fails to propagate to CREB or that there is a mechanistic link between elevated PKA activity and the absence of CREB phosphorylation (Figure 6). In canonical PACAP signalling, PKA activates CREB by phosphorylating Ser133; therefore, a decoupling reaction between PKA and CREB requires additional regulatory mechanisms. The catalytic subunits of PKA can be phosphorylated at specific sites, enhancing the enzymatic activity [41]. Indeed, we demonstrated an elevated level of the phosphorylated form of PKA in both melanoma cell lines upon PACAP application. Activated PKA can phosphorylate transcription factors such as CREB, subsequently enhancing its activity and ability to promote gene expression [42]. In melanoma cells, this activation did not occur, showing the complexity and dual function of PACAP in tumour progression [43]. Another target of PKA could be SOX9 in melanoma cells as it is involved in promoting a cancer stem cell phenotype, contributing to tumour heterogeneity with the crosstalk of SOX10 [44]. In the presence of PACAP, we detected a slight increase in p-SOX9 together with a reduction in SOX10 (Figure 6). In a mouse model, SOX10 and SOX9 act as antagonistic regulators of melanoma development [45]. This shift toward higher SOX9 and lower SOX10 provides a plausible mechanistic explanation for the anti-invasive effects of PACAP observed in our previous studies. SOX10 is well known to promote a more invasive and dedifferentiated melanoma phenotype, in part by sustaining low-MITF, high-motility cell states. In contrast, SOX9 is associated with neural crest specification and when upregulated in melanoma, has been shown to induce a more differentiated and less proliferative phenotype, including cell cycle arrest via p21 [44]. Thus, the PACAP-induced increase in p-SOX9, together with the reduction in SOX10, favours a transcriptional environment that shifts melanoma cells toward a more differentiated and less invasive state. This interpretation is supported by our finding of elevated MITF levels in A2058 and WM35 cells after PACAP treatment (Figure 6), as increased MITF is typically associated with reduced invasiveness and a more melanocytic, less motile phenotype. Taken together, the altered SOX9/SOX10 balance provides a coherent biological link between PACAP stimulation and the reduction in melanoma cell invasiveness, suggesting that PACAP may exert antitumour effects by steering cells away from a SOX10-driven invasive programme and toward a SOX9-mediated differentiated state. SOX9 is a transcription factor that plays a role in neural crest specification and its overexpression in melanoma cell lines has been shown to induce cell cycle arrest through a p21-dependent mechanism [46]. On the other hand, SOX9 is expressed in premigratory neural crest cells and within the pharyngeal apparatus, while SOX10 is present in neural crest cells during their emigration and is crucial for their self-renewal and survival [47]. These results also suggest that SOX regulation via PACAP addition can alter the migration capability of melanoma cells.

The expression of MITF in melanoma is often biphasic; both high and low levels can contribute to melanoma progression through different mechanisms. High MITF levels are associated with increased expression of genes that promote pigmentation and cellular differentiation, whereas low MITF levels favour a more aggressive, invasive phenotype [48]. In melanoma cells, MITF typically translocates to the nucleus, where it binds to DNA and regulates the expression of genes involved in melanocyte differentiation, proliferation, and survival. The subcellular distribution of MITF can therefore markedly influence melanoma cell behaviour. Factors that enhance nuclear localisation tend to promote differentiation and pigmentation, while factors associated with cytoplasmic retention often correlate with a more aggressive phenotype and increased metastatic potential [49]. In the context of our findings, PACAP-induced elevation in MITF—together with increased p-SOX9 and reduced SOX10—suggests a shift toward a more differentiated, less invasive transcriptional state. If PACAP also favours nuclear localisation of MITF, this would further support its antitumour, anti-invasive effect by enhancing MITF-driven differentiation programmes rather than promoting a low-MITF invasive phenotype. In melanoma cells, PACAP addition resulted in a slight increase in MITF expression and a more pronounced immune signal in both of cytoplasm and the nucleus, while melanocytes did not respond in this way. As different melanoma subtypes can exhibit distinct MITF-dependent phenotypes, the biological significance of the observed MITF increase requires careful interpretation. MITF functions in a biphasic manner, where both high and low expression states can contribute to melanoma progression through different mechanisms. In some contexts, elevated MITF promotes differentiation and reduced invasiveness, whereas in others it supports proliferation without necessarily suppressing metastatic potential. Moreover, MITF activity depends not only on its abundance but also on its post-translational modification state and subcellular localisation, which determine whether the protein effectively activates its target genes. Because our current data capture only total MITF levels—and not its nuclear localisation or transcriptional activity—it is difficult to definitively assign the observed increase to either a more differentiated/less invasive programme or a proliferative MITF-high phenotype. Certain melanoma subclasses may rely more heavily on MITF for their growth and survival, while others may downregulate MITF as they progress [48].

Another melanocyte differentiation marker is the DCT which is essential to produce melanin. In melanoma, aberrations in melanocyte function can lead to altered melanin synthesis, potentially influencing the tumour’s characteristics [50]. DCT was shown to interact with components of the tumour microenvironment, affecting not just the cancer cells but also the immune response and the surrounding tissue [51]. Moreover, DCT is not expressed in amelanotic form of the tumour [52]. In the presence of PACAP, increased expression of DCT was detected which can suggest a higher pigmentation process of these cells (Figure 6). The increased DCT expression suggests that PACAP has a balancing function in tumour progression and in melanin production. TYRP1 is another key molecule of melanin synthesis alongside tyrosinase and other enzymes. The expression levels of TYRP1 might have prognostic implications in melanoma. Some studies suggest that higher levels of TYRP1 expression could correlate with more aggressive disease and poorer patient outcomes, while others may indicate a protective effect, highlighting the complexity of its role [53]. TYRP1 is often expressed in melanoma and its increased levels have been found in some metastatic tumours [54]. As a result of PACAP administration, the TYRP1 expression was reduced, demonstrating the complexity of the role of this neuropeptide in melanoma behaviour (Figure 6).

HA is a key component of the tumour microenvironment. High levels of HA can alter the ECM, affecting the behaviour of tumour cells. An increase in HA is often associated with tumour progression, as it can facilitate changes in cell adhesion, migration, and invasion [55]. High levels of HA in the tumour microenvironment may promote the migration and invasion of melanoma cells, contributing to metastatic behaviour [56]. Melanoma cells produce large amounts of HA during early tumourigenesis, whereas at later stages, HA is mainly produced by activated cancer-associated fibroblasts [57]. Interestingly, PACAP enhanced HA production which is contrary to the lowered migration capability of PACAP-treated melanoma cells [29]. HA can promote tumour cell proliferation and survival through interactions with its receptors, primarily CD44 and RHAMM. These interactions can activate signalling pathways that promote cell growth, survival, and proliferation, contributing to tumour progression [15]. In our experiments, CD44 expression did not change significantly in the presence of PACAP [29]. Furthermore, the increased RHAMM expression in the presence of PACAP also suggests an increased metastatic capability (Figure 6) [58]. RHAMM is involved in ECM remodelling, thereby creating a microenvironment that supports tumour growth and metastasis. The interaction between melanoma cells and the ECM components can enable cancer cell dissemination. It is also notable that in melanocytes, these functions are crucial during normal physiological processes such as skin wound healing and pigmentation, where the migration of melanocytes to specific areas is required, thus we suppose that the balancing effect of PACAP on HA receptor expression shifted the melanoma cell signalling pathways toward physiological processes. Furthermore, it has been demonstrated that melanoma progression primarily occurs in a CD44-regulated manner [58]. Importantly, the numerous biological functions of HA are size dependent; therefore, the size of secreted HA could be a key factor influencing metastatic cell migration or regulating intracellular events. However, our study did not include an analysis of HA molecular size, and thus we cannot directly support the size-dependent aspect of HA function in this context. Future investigations that characterise HA size distribution under PACAP treatment will be essential to clarify whether size-specific HA pools contribute to the observed cellular responses.

HA production is primarily mediated by hyaluronan synthase (HAS), enzymes that synthesise HA by catalysing the polymerisation of UDP-glucuronic acid and UDP-N-acetylglucosamine. There are three different HAS—HAS1, HAS2, and HAS3—that synthesise HA in different sizes and with different rates. HAS1 and HAS2 produce high-molecular weight HA (2000 kDa). HAS2 is particularly significant in developmental processes and is often associated with elevated HA levels in pathological conditions, including cancer [23]. The enzymatic activity of various HAS is regulated by their translocation to the plasma membrane from various cytoplasmic organelles, where an intracellular pool of inactive enzymes is present. HAS enzymes function not only for synthesis of HA but also bind this GAG to the cells during the production. In our experiments, increased intracellular and pericellular HA was detected along the elevated expression of HAS2 and HAS3. Upon PACAP action, a strong immune signal was shown in melanoma cells and HAS2 signals were more pronounced than HAS3 signals. Differential regulation of these HAS isoforms may have important functional consequences: shifts toward HAS2-dominated synthesis are expected to favour production of higher-molecular weight HA, which is generally associated with anti-inflammatory, anti-angiogenic and reduced-motility cellular states [59], whereas increased HAS3 activity tends to generate smaller HA fragments that can promote cell proliferation, migration, and pro-tumourigenic signalling [60]. Therefore, changes in the relative expression or activation of HAS2 versus HAS3 could critically influence HA size distribution and in turn, modulate melanoma cell behaviour, including adhesion, motility, and interactions with the tumour microenvironment (Figure 6).

Hyal1 is a lysosomal enzyme that primarily functions to degrade high-molecular weight hyaluronic acid into smaller fragments. Elevated Hyal1 expression has been observed in several cancers, such as prostate, breast and ovarian carcinomas [61,62]. In tumour contexts, increased Hyal1 activity can lead to remodelling of the ECM, and may facilitate tumour cell invasion and migration [63], but in melanoma cells this hyaluronidase was not detected. Hyal2 is also a glycosylphosphatidylinositol-anchored enzyme that initiates the breakdown of extracellular HA into intermediate-sized fragments [64]. It operates mainly at the cell surface in concert with other hyaluronidases. In cancer, Hyal2 has tumour-suppressive functions depending on the context and tumour type [65]. In melanoma, the increased expression by PACAP may indicate a tumour invasion-inhibiting effect of PACAP (Figure 6). There are some experiments which indicate that breast cancer cell lines with greater invasive capabilities tend to have elevated levels of LMW-HA. This increase in LMW-HA correlates with the overexpression of enzymes responsible for its synthesis and degradation such as HAS2 and Hyal2. Importantly, reducing LMW-HA production leads to a marked decrease in the cells’ migratory and invasive behaviours. These results suggest that LMW-HA and the enzymes regulating its metabolism play a significant role in promoting breast cancer invasiveness [66]. In line with these data of the literature, we found elevated HAS2 and Hyal2 expression after PACAP addition, which also suggests a balancing function of PACAP in melanoma motility (Figure 6). Hyal3 is less well-characterised but is believed to have enzymatic activity like other hyaluronidases. Its role in tumours remains less clear, with some evidence suggesting it may modulate HA metabolism locally within the tumour microenvironment [67]. On the other hand, we detected Hyal3 only in melanocytes, in which expression was also promoted by PACAP addition. This result also indicates the HA degradation-inducing effects of PACAP operate independently of the malignancy, but the target hyaluronidase can be different in malignant and normal pigment cells. Hyal4 is a hyaluronidase with activity primarily on chondroitin sulphate rather than HA, but some studies suggest that it may have broader roles in ECM remodelling. Hyal4 expression was weakly detected in melanoma cells and melanocytes and became slightly augmented by PACAP addition. Hyal4’s specific contribution to tumour progression is still under investigation, but it may influence tumour cell interactions with the microenvironment through modulation of glycosaminoglycan composition [68].

Integrin-mediated signalling can promote extracellular matrix remodelling through the upregulation of hyaluronidases, enhancing HA degradation and contributing to tumour aggressiveness [69]. In melanoma, integrin αVβ3 supports adhesion to ECM components such as fibronectin and vitronectin, thereby facilitating cell migration and invasion. It also influences tumour–stroma interactions and can modulate the tumour microenvironment in ways that favour tumour progression [70,71,72]. Although PACAP has been reported to enhance integrin-dependent migration in other cancer models, we detected no changes in αVβ3 integrin subunit expression after PACAP treatment (Figure 6). This indicates that the observed modulation of Hyal expression in our system likely occurs independently of αVβ3-mediated signalling.

In some cancers, PACAP has been observed to promote tumour growth. This can occur via various mechanisms; for instance, PACAP can stimulate cell proliferation by activating signalling pathways such as cAMP-dependent PKA or MAPK [73]. Moreover, PACAP has been shown to stimulate the production of VEGF, which allows tumours to establish their own blood supply, facilitating their growth and metastasis [74]. There is evidence that PACAP might promote cancer cell invasion and metastasis [75]. In particular, PACAP induced activation of certain signalling pathways that might increase the expression of MMPs [37], degrading the extracellular matrix and promoting tumour cell migration. The increased Hyal expression can also affect melanoma progression, although we have detected that PACAP elevated HA production and overall Hyal expression, further suggesting that PACAP acts as a balancing factor in HA metabolism. Conversely, there is also evidence suggesting that PACAP may have anti-cancer effects in some tumour types [76]. PACAP can induce apoptosis in certain tumour cells, especially under specific conditions such as hypoxia [37]. This suggests that PACAP could act as a tumour suppressor under certain circumstances. PACAP has been shown to influence immune responses, which could help the body mount an effective immune defence against tumours [77]. Because HA levels can shape melanoma cell behaviour, these coordinated changes in HAS–Hyal activity suggest a PACAP-driven shift in ECM dynamics rather than a unidirectional pro- or anti-tumour effect.

The role of PACAP in cancer appears to be highly tissue-specific, with PACAP exhibiting both pro-tumourigenic and anti-tumourigenic effects depending on the type of cancer and the stage of tumour progression [73]. In neuroendocrine tumours, e.g., small-cell lung cancer or pheochromocytomas, PACAP might play a role in promoting tumour growth [73,78]. PACAP has been implicated in gliomas, with some studies suggesting that it promotes tumour growth by enhancing cell survival pathways [79]. However, other studies indicate that PACAP’s role in brain tumours is not fully understood and might depend on the tumour’s microenvironment [80]. In some breast and prostate cancer models, PACAP has been suggested to play a role in modulating tumour progression, although the exact effects remain unclear [73]. In melanoma, previously it has been published that PACAP decreased the migration and metastatic formation capability of malignant cells in vitro [29]. Altogether, the role of PACAP in cancer behaviour is controversial, in some cases PACAP expression is associated with more aggressive tumour phenotypes, while in others its expression may correlate with a less aggressive course of the disease. In our previous experiments, PACAP reduced the motility of melanoma cells in vitro, and on the basis of the currently presented data we propose that one of the factors behind this phenomenon can be the balancing effect of PACAP on melanoma HA homeostasis.

Although our findings provide new insight into PACAP-mediated modulation of HA metabolism and melanocytic transcriptional programmes in melanoma, several limitations must be acknowledged. Our analyses were performed in only two melanoma cell lines, which may not capture the full heterogeneity of melanoma subtypes with distinct MITF- or SOX10-dependent phenotypic states. We quantified total HA levels but did not determine HA molecular size distribution. Because HA function is strongly size dependent, the lack of HA size characterisation limits our ability to link specific HA pools to the observed changes in cell behaviour. Moreover, we did not assess downstream functional consequences of altered HAS and Hyal expression beyond HA staining, such as changes in HA secretion kinetics, receptor binding, or ECM mechanical properties. Although PACAP influenced SOX9, SOX10, and MITF levels, we did not measure transcription factor activity, DNA binding, or nuclear–cytoplasmic localisation, making it difficult to determine whether the detected changes translate into functional shifts in transcriptional regulation. Indeed, our study focused on PACAP’s direct effects on melanoma cells and did not include stromal or immune components of the tumour microenvironment, which are known to interact dynamically with HA and may critically shape PACAP’s context-dependent actions. One limitation of this study is the use of GAPDH as the sole housekeeping gene for normalisation. GAPDH expression can vary under certain experimental conditions, including pharmacological treatments, which may introduce bias into the quantification of gene expression. Finally, the experiments relied on short-term PACAP treatments; longer-term or in vivo models could reveal additional regulatory feedback mechanisms that are not captured in the current design.

Together, these limitations indicate that while PACAP appears to function as a context-dependent—but predominantly tumour-suppressive—modulator in melanoma within our experimental conditions, further work incorporating HA size profiling, functional transcription factor assays, and more complex model systems will be essential for fully defining PACAP’s role in melanoma progression.

## 4. Material and Methods

### 4.1. Isolation and Culture of Primary Human Epidermal Melanocytes

Primary human epidermal melanocytes were isolated and cultured according to the protocols of Hajdú et al. [81]. Briefly, samples of juvenile human foreskin (from donors age 2–5) were obtained for melanocyte isolation. The study received ethical approval from the University of Debrecen Ethics Committee (University of Debrecen ETT TUKEB, 25 July 2018, licence number: 5011-2018). After initial sterilisation, tissue samples were incubated overnight in 10 mg/mL dispase II (Gibco, Gaithersburg, MD, USA), dissolved in Hank’s balanced salt solution (Gibco, Gaithersburg, MD, USA) to gently separate the epidermis from the dermis. Subsequently, epidermal tissues were subjected to trypsinisation with 500 µg/mL trypsin/EDTA (Sigma-Aldrich, St. Louis, MO, USA), followed by centrifugation at 450× *g* for 15 min at 4 °C. The resulting cell pellets were resuspended in RPMI-1640 medium (Sigma-Aldrich, St. Louis, MO, USA) and plated into T75 flasks (Eppendorf, Hamburg, Germany). To maintain the primary melanocyte cultures, the RPMI-1640 medium was supplemented with a mitogen cocktail to achieve the following final concentrations: 40 nM TPA (12-O-tetradecanoylphorbol 13-acetate, also known as PMA; Sigma-Aldrich, St. Louis, MO, USA), 40 pM cholera toxin (CT; Sigma-Aldrich, St. Louis, MO, USA), 10 nM endothelin-1 (ET-1; Bachem AG, Bubendorf Switzerland), and 5 ng/mL stem cell factor (SCF; Invitrogen, Carlsbad, CA, USA). The cultured human epidermal melanocytes were routinely examined under light microscopy to verify cell morphology and viability (passage number: 5–9). Cells were maintained at 37 °C in a humidified atmosphere containing 95% air and 5% CO_2_.

### 4.2. Human Melanoma Cell Lines

The human cutaneous melanoma cell line A2058 was established from a lymph node metastasis of an amelanotic melanoma, while WM35 was derived from a non-metastasising human melanoma. Both cell lines were sourced from ATCC (ATCC^®^ CRL-1661™, Manassas, VA, USA). The A2058 melanoma cell line is known for its ability to grow in culture and to express various melanoma-associated markers. The A2058 cell line serves as a model of a highly aggressive metastatic cell population exhibiting intense polymorphism [82]. The WM35 melanoma cell line was established from a patient with early-stage superficial spreading primary melanoma. WM35 cells are characterised as having the capability of metastasis but are not highly aggressive [83]. Both cell lines grow well in culture and express various melanoma-associated markers.

Cells were cultured in RPMI-1640 medium (Sigma-Aldrich, St. Louis, MO, USA) supplemented with 10% foetal bovine serum (Gibco, Gaithersburg, MD, USA), 4.1 g/L glucose, 2 mmol/L L-glutamine (Gibco, Gaithersburg, MD, USA), penicillin (100 units/mL) and streptomycin (100 µg/mL). They were incubated at 37 °C in a humidified environment with 95% air and 5% CO_2_ until they reached approximately 70% confluence in 25 cm^2^ flasks. Additionally, both cell lines were routinely tested every 3rd month for Mycoplasma contamination using polymerase chain reaction (PCR).

### 4.3. Administration of PACAP

PACAP 1-38, at a concentration of 100 nM (from a stock solution of 100 μM, dissolved in sterile distilled water), was used in our experiments. Untreated (control) and PACAP-treated groups were established. Treatment was initiated when cell cultures reached 50–60% confluence and lasted for 48 h. For the PACAP-treated group, the medium was replaced with fresh medium containing 100 nM PACAP after 24 h.

### 4.4. RNA Isolation and Semi-Quantitative Reverse Transcription PCR (RT-PCR)

Melanocyte and melanoma cell cultures were washed three times (melanocytes with DPBS, melanoma cells with PBS). After the final wash was removed, cells were lysed by adding Trizol (Applied Biosystems, Foster City, CA, USA), and after the addition of 20% RNase-free chloroform (Sigma-Aldrich, St. Louis, MO, USA), samples were centrifuged at 10,000× *g* for 15 min at 4 °C. Samples were incubated in 500 µL RNase-free isopropanol at –20 °C for 1 h to precipitate RNA. The RNA was pelleted by centrifugation at 12,000× *g* for 15 min at 4 °C. The pellet was then washed with 75% ethanol, air-dried, and dissolved in nuclease-free water (Promega, Madison, WI, USA) and stored at –70 °C.

The assay mixture for reverse transcriptase reaction contained 2 µg RNA, 0.112 µM oligo(dT), 0.5 mM dNTP, 200 units of High-Capacity RT (Applied Bio-Systems, Foster City, CA, USA) in 1× RT buffer. cDNA was synthesised at 37 °C for 2 h. Amplifications of specific cDNA sequences were performed using primer pairs designed with Primer Premier 5.0 software (Premier Biosoft, Palo Alto, CA, USA), based on human nucleotide sequences available in GenBank. The primers were purchased from Integrated DNA Technologies, Inc. (IDT; Coralville, IA, USA). The specificity of the custom-designed primer pairs was confirmed in silico using the Primer-BLAST service from NCBI (http://www.ncbi.nlm.nih.gov/tools/primer-blast/ 20 July 2018). The nucleotide sequences of the forward and reverse primers, along with the reaction conditions, are detailed in Table 1. The amplifications were conducted in a programmable thermal cycler (Labnet MultiGene™ 96-well Gradient Thermal Cycler; Labnet International, Edison, NJ, USA) with a final reaction volume of 11 μL containing 0.5 μL each of forward and reverse primers (0.4 μM), 0.5 μL of dNTP (200 μM), and 0.625 unit (0.125 μL) of GoTaq^®^ DNA polymerase in 1× Green GoTaq^®^ Reaction Buffer (Promega). The cycling conditions were as follows: initial denaturation at 95 °C for 2 min, followed by 35 cycles of denaturation at 94 °C for 1 min, annealing at the optimised temperatures indicated in Table 1 for 1 min, and extension at 72 °C for 90 s, concluding with a final extension at 72 °C for 10 min. The PCR products were analysed by horizontal electrophoresis on a 1.2% agarose gel containing ethidium bromide (Amresco LLC., Solon, OH, USA) at a constant voltage of 120 V. Amplicon identity was confirmed by using a 100 bp molecular weight ladder to verify the expected amplicon sizes. Signals were visualised using a gel imaging system (Fluorchem E, Protein Simple, San Jose, CA, USA), and the results were normalised to the internal control (GAPDH).

### 4.5. Western Blot Analysis

Cells were washed with physiological NaCl solution and subsequently harvested. After centrifugation, cell pellets were resuspended in 100 μL of RIPA (Radioimmunoprecipitation Assay) homogenisation buffer (Sigma-Aldrich, St. Louis, MO, USA), containing 150 mM NaCl, 1.0% NP40, 0.5% sodium deoxycholate, 50 mM Tris, 0.1% SDS, pH 8.0, supplemented with a protease and phosphatase inhibitor cocktail containing the following (all from Sigma-Aldrich, St. Louis, MO, USA): aprotinin (10 µg/mL), benzamidine (5 mM), leupeptin (10 µg/mL), trypsin inhibitor (10 µg/mL), PMSF (1 mM), EDTA (5 mM), EGTA (1 mM), sodium fluoride (8 mM), and sodium orthovanadate (1 mM). The samples were stored at −70 °C.

To ensure complete lysis and to shear genomic DNA, the suspensions were then sonicated on ice with an ultrasonic homogeniser (Cole-Parmer, Vernon Hills, IL, USA) using a pulsed protocol. Protein concentration of the lysates was determined using a BCA protein assay kit (Thermo Fisher Scientific, Waltham, MA USA). Samples were denatured by adding Laemmli electrophoresis sample buffer (containing 4% SDS, 10% 2-mercaptoethanol, 20% glycerol, 0.004% bromophenol blue, and 0.125 M Tris–HCl, pH 6.8) and boiling for 10 min at 95 °C.

For Western blotting, 20 µg of protein from each sample was used and proteins were separated by 7.5% SDS-PAGE and transferred to nitrocellulose membranes (Bio-Rad Trans Blot Turbo Midi Nitrocellulose Transfer Packs) using a Bio-Rad Trans-Blot Turbo system (Bio-Rad Laboratories, Hercules, CA, USA). Equal loading was confirmed with Ponceau-staining. The target proteins detected were VPAC1, PAC1, DPP4, PKA, p-PKA, CREB, p-CREB, SOX9, p-SOX9, SOX10, MITF, DCT, TYRP1, HAS1, HAS2, HAS3, CD44, RHAMM, Hyal1, Hyal2, Hyal3, Hyal4, Integrin αV, Integrin β3, and actin (further details in Table 2). The chosen proteins represent key components of the signalling pathways, transcriptional networks, and functional outputs relevant to the study. They allow assessment of receptor presence, downstream signalling activation, lineage-specific transcription factors, melanocytic differentiation markers, and hyaluronan metabolism. After blocking in 5% non-fat dry milk in PBST for 1 h, the membranes were incubated with primary antibodies overnight at 4 °C, as specified in Table 2. Following the primary antibody incubation, the membranes were washed three times for 10 min each in PBST and were then incubated with horseradish peroxidase-conjugated secondary antibodies, anti-rabbit IgG (Bio-Rad Laboratories, Hercules, CA, USA) or anti-mouse IgG (Bio-Rad Laboratories, Hercules, CA, USA), both at a dilution of 1:1500. Signals were developed using enhanced chemiluminescence (Advansta Inc., Menlo Park, CA, USA) according to the manufacturer’s instructions. The gel imaging system (Fluorchem E, Protein Simple, San Jose, CA, USA) captured the signals, and the results were normalised to actin expression.

### 4.6. Immunocytochemistry

Immunocytochemistry was conducted on cells grown on coverslips to visualise the intracellular localisation of DPP4, MITF, RHAMM, HAS2, and HAS3. The cultures were fixed for 1 h in a solution of 4% paraformaldehyde (Sigma-Aldrich, St. Louis, MO, USA) and then washed with distilled water. Following rinsing in PBS (pH 7.4), cells were permeabilised and blocked for 30 min at 37 °C in PBST (PBS containing 1% BSA and 0.1% Tween-20 [Amresco LLC., Solon, OH, USA]).The samples were incubated overnight at 4 °C with rabbit polyclonal anti-DPP4 antibody (Cell Signaling, Danvers, MA, USA) at 1:500 dilution, rabbit polyclonal anti-MITF antibody (Abcam, Cambridge, UK) at 1:500 dilution, rabbit polyclonal anti-RHAMM antibody (Novocastra, Newcastle, UK) at 1:400 dilution, rabbit polyclonal anti-HAS2 antibody (Santa Cruz Biotechnology, Inc., Santa Cruz, CA, USA) at 1:200 dilution, and rabbit polyclonal anti-HAS3 antibody (Abcam, Cambridge, UK) at 1:500 dilution in PBST. Subsequently, the primary antibody was visualised using an anti-rabbit Alexa Fluor 555 secondary antibody (Life Technologies Corporation, Carlsbad, CA, USA) at a 1:1000 dilution in PBST, followed by three washes in PBS. The cultures were then mounted in Vectashield Hard Set mounting medium (Vector Laboratories, Burlingame, CA, USA) containing DAPI for nuclear staining. Each reaction was performed in triplicate and five distinct visual fields were examined per sample.

An Olympus FV3000 confocal microscope (Olympus Corporation, Tokyo, Japan), equipped with a 60× Plan Apo N oil immersion objective (NA: 1.42) and controlled by FV31S-SW software (Ver.2.6) (Olympus Corporation, Tokyo, Japan), captured fluorescent images to assess the subcellular localisation of DPP4, MITF, RHAMM, HAS2, and HAS3. Z-series images, with a 1 μm optical thickness, were recorded in sequential scan mode using laser beams with excitation wavelengths of 405 nm (for DAPI) and 543 nm (for Alexa Fluor 555). The average pixel time was 4 µs. Images from the Alexa 555 and DAPI channels were superimposed using Adobe Photoshop software 10.0. Images were used to assess the presence, localisation, and relative intensity of the detected proteins. No quantitative image analysis was applied and interpretations were based solely on visual comparison across conditions.

### 4.7. Hyaluronic Acid Detection

To detect hyaluronic acid (HA), an affinity cytochemistry method was employed. Melanoma cell cultures were stained using a biotinylated HA-binding complex (bHABC). The cultures were fixed for 1 h in a solution of 4% paraformaldehyde (Sigma-Aldrich, St. Louis, MO, USA) and then washed with distilled water. Following rinsing in PBS (pH 7.4), nonspecific binding sites were blocked by incubating the samples with PBST containing 1% BSA (Amresco LLC., Solon, OH, USA) for 30 min at 37 °C. A biotinylated HA-binding complex at a concentration of 5 µg/mL (bHABC was generously provided by R. Tammi and M. Tammi from the Department of Anatomy, University of Kuopio, Kuopio, Finland) was used to detect HA, and samples were incubated overnight at 4 °C. The reaction was visualised with Streptavidin-Alexa 555 (2 µg/mL, Invitrogen, Thermo Fisher Scientific, Carlsbad, CA, USA) for fluorescence microscopy. Cell cultures were mounted in Vectashield Hard Set mounting medium (Vector Laboratories, Burlingame, CA, USA) containing DAPI for nuclear staining. For quantification, all images were acquired using identical exposure settings. Red pixel intensity was measured for each slide, and quantification criteria were applied using standardised thresholds for signal detection (integrated density). Images were analysed using the same parameters across all samples to ensure valid comparison between experimental groups.

### 4.8. Statistical Analysis

Data from RT-PCR and Western blot analyses are presented as mean values ± standard deviation (SD) from at least three independent experiments. Statistical significance between untreated (control) and PACAP 1-38-treated cells was determined by Student’s *t* test (GraphPad Prism 10). A *p*-value of less than 0.05 was considered statistically significant.

## 5. Conclusions

In melanoma, PACAP has been reported to reduce cell migration and metastatic potential in vitro [27], suggesting a tumour-suppressive influence. Our previous findings align with this, as PACAP decreased melanoma cell motility. The current data further indicate that PACAP exerts a balancing effect on HA homeostasis by simultaneously increasing HA production and elevating Hyal expression. This coordinated regulation of HA turnover may contribute to a less migratory phenotype by stabilising ECM dynamics rather than driving aggressive remodelling. Taken together, the available evidence supports the view that PACAP acts as a context-dependent modulator in melanoma, but with predominantly tumour-suppressive effects under the conditions examined, particularly through its impact on HA metabolism.

## Figures and Tables

**Figure 1 ijms-26-12049-f001:**
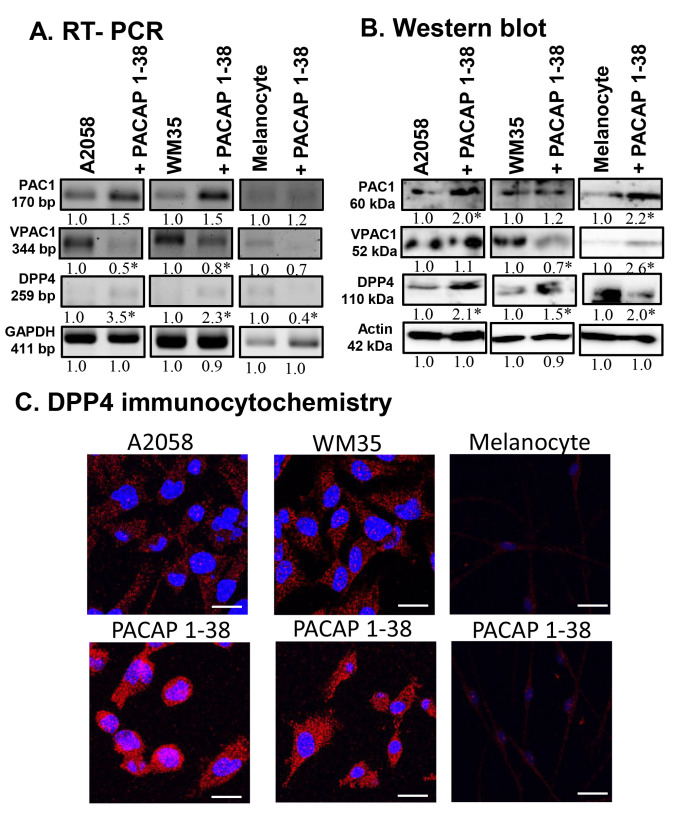
mRNA (**A**) and protein (**B**) expression of PACAP receptors and DPP4 in human melanoma cells and in melanocytes. For RT-PCR and for Western blots, GAPDH and actin were used as internal controls, respectively. Images represent data from three independent experiments. Optical density of signals was measured, and the results were normalised to the optical density of control cultures. For panels (**A**,**B**), numbers below signals represent integrated densities of signals determined using ImageJ 1.40 g software. All data are the average of at least three different experiments. Statistical analysis was performed using Student’s *t* test. All data were normalised to GAPDH and actin. Data are expressed as the mean ± SD. Asterisks (**A**,**B**) indicate significant (* *p* < 0.05) alterations in the PACAP 1-38 treatment group as compared to the control. (**C**) Immunocytochemical reactions of DPP4 in melanoma and melanocytes. Magnification of the images is 60×. Scale bar: 5 µm.

**Figure 2 ijms-26-12049-f002:**
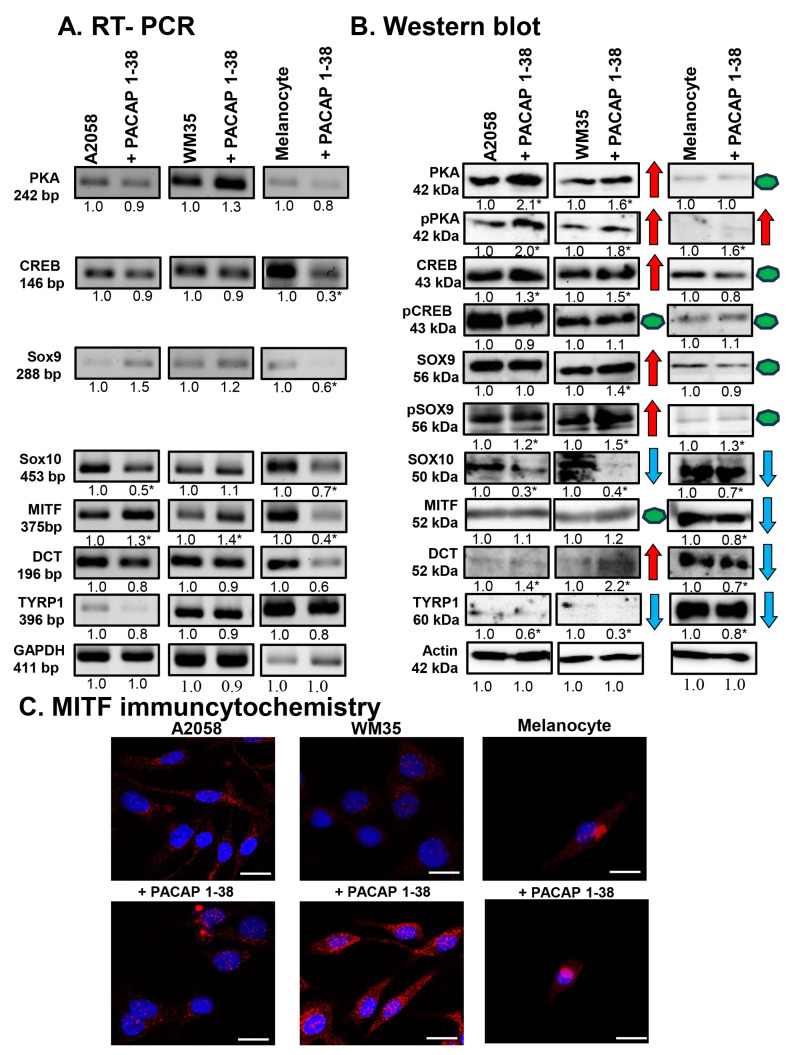
mRNA (**A**) and protein (**B**) expression of PKA, CREB, SOX9, SOX10, MITF, DCT, and TYRP1 in A2058, WM35 cells, and epidermal melanocytes. For RT-PCR and for Western blots, GAPDH and actin were used as internal controls, respectively. Images represent data from three independent experiments. Optical density of signals was measured, and the results were normalised to the optical density of control cultures. For panels (**A**,**B**), numbers below signals represent integrated densities of signals determined using ImageJ software. All data are the average of at least three different experiments. Statistical analysis was performed using Student’s *t* test. All data were normalised to GAPDH and actin. Representative blots from three independent experiments are shown. Asterisks (**A**,**B**) indicate significant (* *p* < 0.05) alterations in the PACAP 1-38 treatment group as compared to the control. For panels in (**B**), red arrows indicate upregulated molecules or processes, while blue arrows represent downregulation, and green dots indicate no alteration in protein expression. (**C**) Immunocytochemical reactions of MITF in melanoma cells and melanocytes. Magnification of the images is 60×. Scale bar: 5 µm.

**Figure 3 ijms-26-12049-f003:**
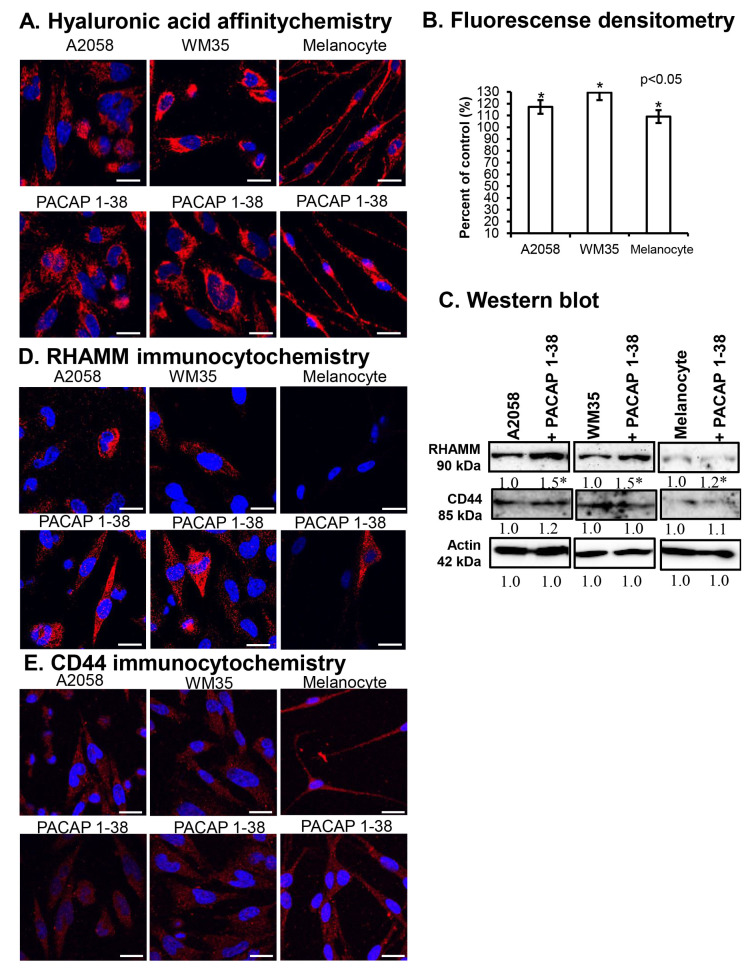
HA production of cells (**A**) and the densitometric analysis of HA (**B**). Protein (**C**) expression of RHAMM and CD44 in A2058, WM35 cells, and epidermal melanocytes. For Western blots, actin was used as internal control, respectively. Images represent data from three independent experiments. Optical density of signals was measured, and the results were normalised to the optical density of control cultures. For panels in (**C**), numbers below signals represent integrated densities of signals determined using ImageJ 1.40 g software. All data are the average of at least three different experiments. Statistical analysis was performed using Student’s *t* test. All data were normalised to actin. Data are expressed as the mean ± SD. Asterisks (**A**,**D**) indicate significant (* *p* < 0.05) alterations in the PACAP 1-38 treatment group as compared to the control. (**D**,**E**) Immunocytochemical reactions of RHAMM and CD44 in melanoma cells and melanocytes. Magnification of the images is 60×. Scale bar: 5 µm.

**Figure 4 ijms-26-12049-f004:**
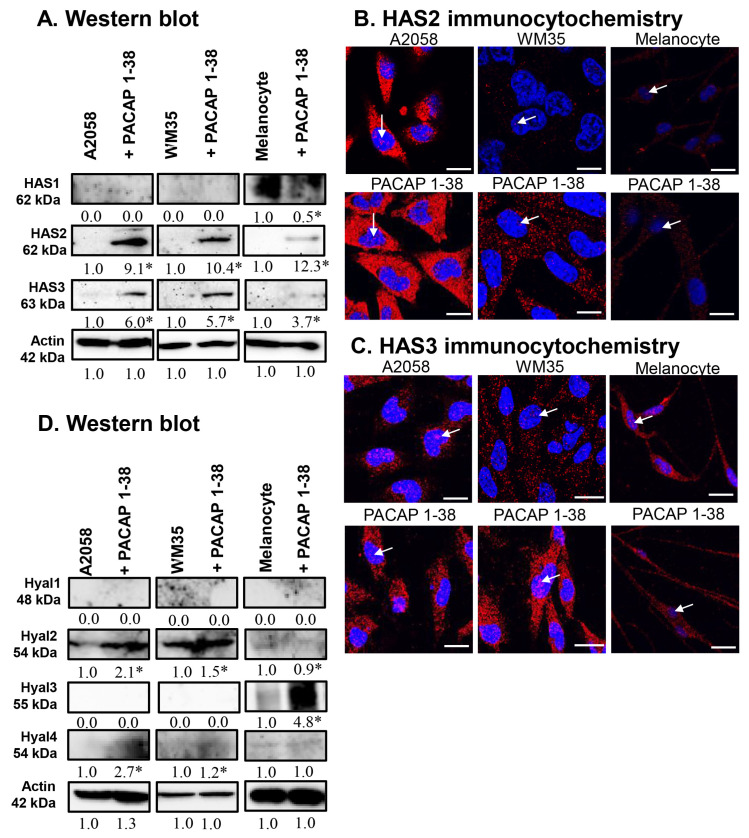
Protein (**A**) expression of HAS1, HAS2, and HAS3 in A2058, WM35 cells, and epidermal melanocytes. For Western blots, actin was used as internal control, respectively. Images represent data from three independent experiments. (**B**,**C**) Immunocytochemical reactions of HAS2 and HAS3 in melanoma cells and melanocytes. White arrows indicate the nuclear localization of HAS2 and HAS3. Magnification of the images is 60×. Scale bar: 5 µm. Protein (**D**) expression of Hyal1, Hyal2, Hyal3, and Hyal4 in A2058, WM35 cells, and epidermal melanocytes. For Western blots, actin was used as internal control. Images represent data from three independent experiments. Optical density of signals was measured, and the results were normalised to the optical density of control cultures. For panels (**A**,**D**), numbers below signals represent integrated densities of signals determined using ImageJ software. All data are the average of at least three different experiments. Statistical analysis was performed using Student’s *t* test. All data were normalised to actin. Data are expressed as the mean ± SD. Asterisks (**A**,**D**) indicate significant (* *p* < 0.05) alterations in the PACAP 1-38 treatment group as compared to the control.

**Figure 5 ijms-26-12049-f005:**
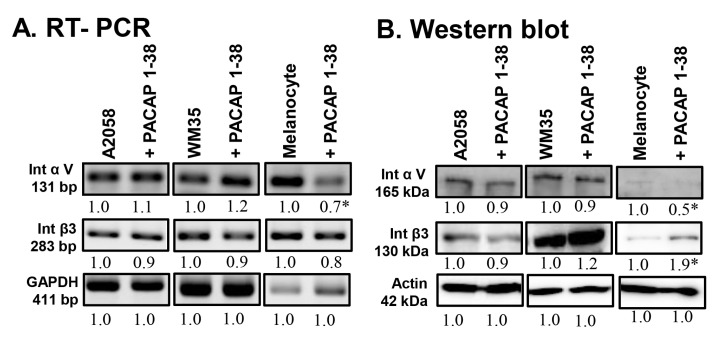
mRNA (**A**) and protein (**B**) expression of integrin αV and integrin β3 in human melanoma cells and melanocytes. For RT-PCR and for Western blots, GAPDH and actin were used as internal controls, respectively. Images represent data from three independent experiments. Optical density of signals was measured, and the results were normalised to the optical density of control cultures. For panels (**A**,**B**), numbers below signals represent integrated densities of signals determined using ImageJ software. All data are the average of at least three different experiments. Statistical analysis was performed using Student’s *t* test. All data were normalised to actin. Data are expressed as the mean ± SD. Asterisks (**A**,**B**) indicate significant (* *p* < 0.05) alterations in the PACAP 1-38 treatment group as compared to the control.

**Figure 6 ijms-26-12049-f006:**
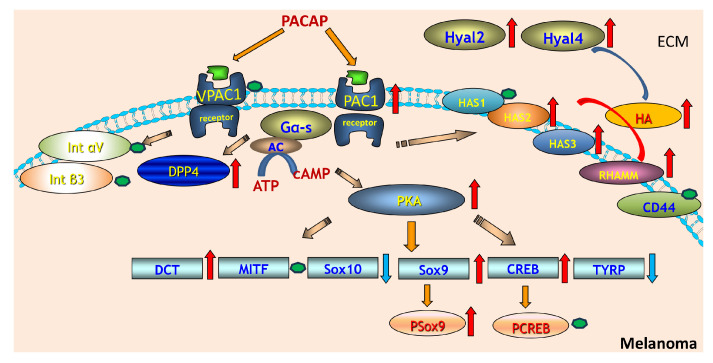
PACAP-driven signalling cascade regulating hyaluronan metabolism and melanocytic gene expression in melanoma. PACAP activates VPAC1 and PAC1 receptors on melanoma cells, triggering Gαs–adenylate cyclase signalling and increased cAMP production. Elevated cAMP activates PKA, which subsequently enhances phosphorylation of downstream transcription factors including CREB and Sox9, while modulating Sox10 and TYRP expression. PACAP signalling also upregulates DCT and MITF, key regulators of melanocyte differentiation. On the extracellular side, PACAP stimulation increases the expression of hyaluronan synthases (HAS2–3) and hyaluronidases (Hyal2, Hyal4), leading to elevated hyaluronan (HA) synthesis and turnover. HA interacts with receptors such as RHAMM and CD44, influencing extracellular matrix remodelling and cell signalling. Integrins (αV and β3) and DPP4 are shown as additional surface components affected by PACAP-mediated pathways. Red arrows indicate upregulated molecules or processes, while blue arrows represent downregulation, while green dots indicate no alterations in protein expression of these proteins.

**Table 1 ijms-26-12049-t001:** Table of primer pairs and PCR conditions.

Gene	Primer	Nucleotide Sequence (5’→3’)	GenBank ID	Annealing Temperature	Amplimer Size (bp)
*CREB*	sense antisense	AGC CGA GAA CCA GCA GAG (269–286) GAG TTA CGG TGG GAG CAG (397–414)	**NM_001371427.1**	53 °C	146
*DCT*	sense antisense	GCG GGG CTG GAG AAA CGA (126–143) AGA AGT GGG CGC TGT GGC (304–321)	**NM_001129889.3**	58 °C	196
*DPP4*	sense	GGA AGT CAT CGG GAT AG (1767–1783)	**NM_001935.4**	50 °C	259
	antisense	ATC ATT CAC GCT GCT GT 2025–2009)			
*Int* *α* *V*	sense	GTG GAC AGT CCT GCC GAG TA (386–405) GGC TGG GTG GTG TTT GCT (499–516)	**NM_002210.5**	56 °C	131
*Int β3*	sense	CGT GCT GAC GCT AAC TGA (674–691) GGT AGT GGA GGC AGA GTA ATG (936–956)	**NM_000212.3**	53 °C	283
*MITF*	sense	ATT CCA TCC ACG GGT CTC (1364–1381) AGG GAG GAT TCG CTA ACA (1721–1738)	**NM_198159.3**	51 °C	375
*PAC1*	sense	TCA TCC TTT GTC GCT TCC (807–824)	**NM_001199637.2**	53 °C	170
	antisense	GAC GGC CTT ACA TTC CAC (976–959)			
*Sox9*	sense	TCA AAG GCT ACG ACT GGA CG (1364–1381) GCG GCT GGT ACT TGT AATC C (882–901)	**NM_000346.4**	56 °C	288
*Sox10*	sense antisense	GCT GAA CGA AAG TGA CAA GC (710–729) GCC ACA TCA AAG GTC TCC AT (1143–1162)	**NM_006941.4**	58 °C	453
*TYRP1*	sense antisense	ATG TCG CTC AGT GCT TG (995–1011) AAG TGT ATC CCA GGT TGT C (1372–1390)	**NM_000550.3**	56 °C	396
*VPAC1*	antisense	CCA TTG CCT GTG GTT TG (857–873)	**NM_004624.4**	54 °C	344
	antisense	CAG CCA GAA GAA GTT AGC C (1200–1182)			
*GAPDH*	sense	CCA GAA GAC TGT GGA TGG CC (740–759)	**NM_002046.5**	54 °C	411
	antisense	CTG TAG CCA AAT TCG TTG TC (1150–1131)			

**Table 2 ijms-26-12049-t002:** Table of antibodies used in the experiments.

Antibody	Host Animal	Dilution	Cat. Num	Distributor
Anti-PAC1	rabbit, polyclonal	1:500	SAB2900693	Sigma-Aldrich, St. Louis, MO, USA
Anti-VPAC1	rabbit, polyclonal	1:800	AVR-001	Alomone, Jerusalem, Israel.
Anti-DPP4	rabbit, polyclonal	1:500	61408	Cell Signaling, Danvers, MA, USA
Anti-PKA	rabbit, polyclonal	1:800	4782	Cell Signaling, Danvers, MA, USA
Anti-p-PKA	rabbit, polyclonal	1:500	4781	Cell Signaling, Danvers, MA, USA
Anti-CREB	rabbit, polyclonal	1:800	06-863	Millipore, Burlington, MA, USA
Anti-p-CREB	rabbit, polyclonal	1:800	06-519	Millipore, Burlington, MA, USA
Anti-SOX9	rabbit, polyclonal	1:500	ab3697	Abcam, Cambridge, UK
Anti-p-SOX9	rabbit, polyclonal	1:800	SAB4503991	Sigma-Aldrich, St. Louis, MO, USA
Anti-SOX10	rabbit, polyclonal	1:500	ab107532	Abcam, Cambridge, UK
Anti-MITF	rabbit, polyclonal	1:800	ab20663	Abcam, Cambridge, UK
Anti-DCT	rabbit, polyclonal	1:800	ab115785	Abcam, Cambridge, UK
Anti-TYRP1	rabbit, polyclonal	1:800	HPA000937	Sigma-Aldrich, St. Louis, MO, USA
Anti-RHAMM	mouse, monoclonal	1:500	NCL-CD168	Novocastra, Newcastle, UK
Anti-CD44	mouse, monoclonal	1:800	BBA10	R&D Systems, Minneapolis, MN, USA
Anti-HAS1	goat, polyclonal	1:200	SAB4300848	Sigma-Aldrich, St. Louis, MO, USA
Anti-HAS2	rabbit, polyclonal	1:200	sc-66916	Santa Cruz Biotechnology Inc., Santa Cruz, CA, USA
Anti-HAS3	rabbit, polyclonal	1:400	ab170872	Abcam, Cambridge, UK
Anti-Hyal1	rabbit, polyclonal	1:400	HPA002112	Sigma-Aldrich, St. Louis, MO, USA
Anti-Hyal2	rabbit, polyclonal	1:400	SAB1100696	Sigma-Aldrich, St. Louis, MO, USA
Anti-Hyal3	rabbit, polyclonal	1:400	SAB1406655	Sigma-Aldrich, St. Louis, MO, USA
Anti-Hyal4	rabbit, polyclonal	1:400	A96606	Sigma-Aldrich, St. Louis, MO, USA
Anti-integrin αV	rabbit, polyclonal	1:400	4711	Cell Signaling, Danvers, MA, USA
Anti-integrin β3	rabbit, polyclonal	1:800	06-863	Cell Signaling, Danvers, MA, USA
Anti-Actin	mouse, monoclonal	1:10,000	06-519	Sigma-Aldrich, St. Louis, MO, USA
Anti-Rabbit IgG HRP conjugate	goat	1:1500	170-6515	Bio-Rad Laboratories, Hercules, CA, USA
Anti-Mouse IgG HRP conjugate	goat	1:1500	170-6516	Bio-Rad Laboratories, Hercules, CA, USA

## Data Availability

The original contributions presented in this study are included in the article. Further inquiries can be directed to the corresponding author.

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
