# Peer review of "Exploring the Link Between PACAP Signalling and Hyaluronic Acid Production in Melanoma Progression"

_ijms, 2025, doi:10.3390/ijms262412049_

Round 1

Reviewer 1 Report

Comments and Suggestions for Authors

Reviewer Comments to the Authors

General Overview

This study examined the effects of pituitary adenylate cyclase activating polypeptide (PACAP) on A2058 and WM35 melanoma cell lines. The authors indicated that PACAP modulates several molecules linked to melanocyte differentiation and extracellular matrix composition. Importantly, PACAP increased SOX9 but decreased SOX10 expression. PACAP had no effect on CREB signaling, but it lowered TYRP1, raised DCT levels, and altered MITF expression and localization. It also enhanced hyaluronan (HA) production, upregulated HA synthases, RHAMM, and hyaluronidase 2 (Hyal2), while leaving αV and β3 integrin expression unchanged. These findings indicate that PACAP influences hyaluronan homeostasis and tumor matrix composition. The authors indicated that PACAP can potentially affect melanoma cell invasiveness.

INTRODUCTION

This section provided information in general and broad terms. The information was not linked to the main theme of the manuscript, “Melanoma.” This section should be revised to focus on melanoma. I have the following suggestions for the authors:

  • Line 47–49: Please provide clarification about the link between embryonic melanocyte motility and metastatic potential.
  • Line 53–55: The authors described UV radiation as the “primary” risk factor in melanoma. However, genetic and other environmental factors also play significant roles in the pathogenesis of this fatal neoplasm. Please mention the interaction of UV with genetic predisposition.
  • Line 64–67: The statement about signalling pathways is general and lacks specific context relevant to melanoma. The authors need to rewrite this sentence to include key melanoma-related pathways.
  • Line 69–79: The section on hyaluronic acid (HA) is broad. The authors should rewrite the sentence connecting HA function directly to melanoma invasion or metastasis.
  • Line 88–91: The description of hyaluronidases (HYALs) is broad. The authors should rewrite this sentence to indicate the specific roles of HYALs in melanoma progression or invasion.
  • Line 93–115: PACAP biology is explained in broad terms. The authors should rewrite this section to indicate how these pathways may influence melanoma progression and invasion.
  • Line 137–138: Please provide a short background comment introducing readers to the functions of SOX9, SOX10, and MITF. For instance, SOX10 promotes melanocyte lineage commitment, and MITF regulates pigmentation and survival of melanocytes.

MATERIALS AND METHODS

I have the following suggestions for the authors:

2.1. Isolation and Culture of Primary Human Epidermal Melanocytes

  • Line 143–144: Please indicate the donor age range and number of samples.
  • Line 146–150: Please justify Dispase and trypsin enzyme concentrations used in the investigation.
  • Line 153–156: For the listed mitogen cocktail composition, please mention the rationale for each factor.
  • Line 158–159: Please mention the passage number of melanocytes at the time of experiments.

2.2. Human Melanoma Cell Lines

  • Line 176: Please indicate how often Mycoplasma testing was performed (e.g., monthly).

2.3. Administration of PACAP

  • Line 178–183: Please provide rationales for (i) the PACAP concentration (100 nM) and (ii) exposure time (48 h).

2.4. RNA Isolation and Semi-Quantitative RT-PCR

  • Line 193–201: Please include amplicon sizes and confirmation method (gel size or melting curve). If there is no confirmation method, then acknowledge this as a limitation of your study (Discussion section).
  • Line 210–213: GAPDH is used as the sole housekeeping gene, which may vary under treatment. Please acknowledge this as a limitation of your study (Discussion section).

2.5. Western Blot Analysis

  • Line 217–225: Please mention how equal loading was confirmed (e.g., Ponceau S staining).
  • Line 236–239: Please indicate why these specific markers (target proteins) were chosen.

2.6. Immunocytochemistry

  • Line 266–267: Please indicate whether imaging was qualitative or if quantification criteria were applied.

2.7. Hyaluronic Acid Detection

  • Line 277–286: The method describes staining but not quantification. These qualitative results may not reflect total HA content. Please acknowledge this as a limitation of your study (Discussion section).

RESULTS

I am happy with this section as it stands. I have no suggestions for the authors.

DISCUSSION

The current Discussion section, as it stands, merely reiterates the general literature already covered in the Introduction section. The authors did not offer the reader sufficient critical analysis of the study’s own findings.

The authors should rewrite the Discussion section as follows:
First, clearly summarize the key and novel findings of this investigation. Following this, the Discussion should be organized into distinct subheadings, each dedicated to one major finding. Within each subsection, the authors should critically analyze that specific result in the context of existing knowledge. For each finding, the authors should systematically highlight its significance, possible underlying molecular mechanisms, and potential therapeutic implications (if applicable). In short, the authors should focus only on how current findings improve our previous knowledge about PACAP in melanoma. Also, I have the following suggestions for the authors:

  • Line 476–486: Please directly relate the reported increase in HA to your own data and clarify whether PACAP’s effects are pro- or anti-invasive in this context.
  • Line 493–498: The authors indicated a mechanistic link between PKA phosphorylation and lack of CREB activation. Please provide evidence for this link.
  • Line 509–515: Please connect how the altered SOX9/SOX10 balance could explain the observed PACAP effects on invasiveness.
  • Line 517–526: Please speculate on why PACAP affects MITF localization/distribution.
  • Line 529–533: The statement that the significance of MITF change is “difficult to judge” needs a reasoned interpretation from the authors.
  • Line 573–575: The authors indicated the size-dependence of HA effects, but this claim is not supported by data. Please acknowledge the lack of HA size analysis and recommend it for future work.
  • Line 585–588: Explain how differential HAS isoform regulation could influence HA size and tumor cell behavior.
  • Line 620–634: The discussion of integrins is extensive but does not connect back to experimental results showing no changes. Please shorten this part of the discussion.
  • Line 636–657: Please limit the discussion to mechanisms relevant to current findings (HA, HAS, Hyal, SOX9/10, MITF).
  • Line 658–671: Please focus on melanoma, with a clear statement on whether PACAP acts overall as a tumor-suppressive or context-dependent modulator in melanoma.
  • Line 701: Please add a section addressing the limitations of your current investigation.

Author Response

This section provided information in general and broad terms. The information was not linked to the main theme of the manuscript, “Melanoma.” This section should be revised to focus on melanoma. I have the following suggestions for the authors:. 

Thank you for your feedback regarding the manuscript. To enhance the Introduction, we revised the chapter to more explicitly define the knowledge of other laboratories for focusing on melanoma.

Line 47–49: Please provide clarification about the link between embryonic melanocyte motility and metastatic potential.

Melanocytes differentiate early in development from melanoblasts arising from neural crest cells —a transient, highly migratory, stem-like population—and these melanoblasts reach their destinations in the skin, eye, and other tissues by using EMT-like programs, dynamic cytoskeletal remodeling, high motility and invasiveness through extracellular matrices and chemotactic cues such as endothelin, Wnt, and KIT ligand. Melanoma cells exploit and reactivate these same neural-crest-derived migratory programs by switching from a proliferative MITF-high state to an invasive MITF-low state, a transition that drives migration through extracellular matrix, intravasation/extravasation and ultimately early metastatic spread.

Line 53–55: The authors described UV radiation as the “primary” risk factor in melanoma. However, genetic and other environmental factors also play significant roles in the pathogenesis of this fatal neoplasm. Please mention the interaction of UV with genetic predisposition.

One important interaction is that UV radiation can have a stronger carcinogenic effect in individuals with inherited susceptibility—such as pathogenic variants in CDKN2A, MC1R, or other genes involved in pigmentation and DNA repair—which increase sensitivity to UV-induced DNA damage and reduce the efficiency of its repair, thereby amplifying melanoma risk beyond the effect of UV exposure alone.

Line 64–67: The statement about signalling pathways is general and lacks specific context relevant to melanoma. The authors need to rewrite this sentence to include key melanoma-related pathways.

Signalling molecules within the melanoma tumour microenvironment, such as those involving the MAPK, PI3K/AKT, and Wnt pathways, can profoundly influence tumour behaviour and melanoma cells actively manipulate these pathways to remodel their microenvironment, promoting angiogenesis, sustaining growth and evading immune detection.

Line 69–79: The section on hyaluronic acid (HA) is broad. The authors should rewrite the sentence connecting HA function directly to melanoma invasion or metastasis.

Thank you for your suggestions. We have rewritten the chapter focusing on melanoma.

Line 88–91: The description of hyaluronidases (HYALs) is broad. The authors should rewrite this sentence to indicate the specific roles of HYALs in melanoma progression or invasion.

Hyaluronidases (HYAL1–4), the enzymes responsible for degrading HA, contribute to melanoma progression by generating low-molecular-weight HA fragments that enhance melanoma cell motility, invasion, and metastatic potential, and by modulating CD44- and RHAMM-mediated signalling pathways that support a pro-tumorigenic microenvironment.

Line 93–115: PACAP biology is explained in broad terms. The authors should rewrite this section to indicate how these pathways may influence melanoma progression and invasion.

Pituitary adenylate cyclase–activating polypeptide (PACAP) is a 38-amino-acid neuropeptide structurally related to VIP and best known for roles in neuroprotection, neurodevelopment, and immune modulation. PACAP signals primarily through the high-affinity PAC1 receptor and, to a lesser extent, through VPAC1 and VPAC2, activating intracellular pathways such as adenylate cyclase/cAMP–PKA, PLC–IP3/DAG–PKC, and multiple MAPK cascades (ERK, JNK, p38). These same pathways are critically involved in melanoma cell survival, proliferation, plasticity, and motility. Emerging evidence suggests that PACAP–PAC1 signalling may influence melanoma progression by enhancing MAPK and cAMP-driven transcriptional programs that support tumour growth, by modulating cytoskeletal dynamics through PKC and calcium signalling, and by interacting with immune-regulatory mechanisms that can facilitate tumour invasion and immune evasion. Consequently, dysregulated PACAP signalling within the melanoma microenvironment may contribute to increased aggressiveness and metastatic behaviour.

Line 137–138: Please provide a short background comment introducing readers to the functions of SOX9, SOX10 and MITF. For instance, SOX10 promotes melanocyte lineage commitment, and MITF regulates pigmentation and survival of melanocytes.

SOX9, SOX10, and MITF are key transcription factors in melanocyte development and function. SOX10 promotes neural crest–derived melanocyte lineage commitment and drives the expression of melanocyte-specific genes, while SOX9 contributes to melanocyte differentiation and stress responses. MITF acts as a master regulator of melanocyte biology, controlling pigmentation, survival, proliferation, and cellular plasticity, and its activity integrates signals from multiple developmental and oncogenic pathways.

Line 143–144: Please indicate the donor age range and number of samples.

We analyzed samples obtained from donors aged 2–5 years.

Line 146–150: Please justify Dispase and trypsin enzyme concentrations used in the investigation.

We used 10 mg/mL Dispase II and 500 µg/mL trypsin/EDTA based on established protocols for efficient and reproducible epithelial–mesenchymal separation while preserving cell viability and tissue architecture. Dispase at 10 mg/mL provides sufficient proteolytic activity to gently dissociate extracellular matrix components without causing excessive epithelial damage, which is critical for isolating intact cell layers. The use of 500 µg/mL trypsin/EDTA provides a standard, mild secondary dissociation step that effectively releases individual cells while minimizing over-digestion. These concentrations are widely used in tissue dissociation workflows and were selected to balance dissociation efficiency with the preservation of cellular integrity and downstream assay performance.

Line 153–156: For the listed mitogen cocktail composition, please mention the rationale for each factor.

The rationale for the individual components of the mitogen cocktail is to ensure the selective proliferation and optimal maintenance of the primary human epidermal melanocyte cultures [81]. This cocktail is based on established protocols to mimic the necessary growth environment, partially replacing the factors normally provided by adjacent keratinocytes in vivo.

The function of each component is as follows:

-           TPA (12-O-tetradecanoylphorbol 13-acetate, also known as PMA) is a potent, selective mitogen that activates the Protein Kinase C (PKC) signaling pathway [Yamasaki T et al (2009) Phosphorylation of 763activation transcription Factor-2 at serine 121 by protein kinase C controls c-Jun-mediated activation of transcription. J Biol Chem 284(13):8567–8581. https://doi.org/10.1074/jbc.M808719200]. This activation subsequently leads to the phosphorylation of several transcription factors, which is essential for initiating and sustaining the proliferation and differentiation of the cultured melanocytes.

-           Cholera Toxin (CT) acts as a cAMP-inducer, mimicking the effects of the melanocyte-stimulating hormone (MSH) [O’Keefe E, Cuatrecasas P (1974) Cholera toxin mimics melanocyte stimulating hormone in inducing differentiation in melanoma cells. Proc Natl Acad Sci USA 71(6):2500–2504. https://doi.org/10.1073/pnas.71.6.2500]. CT specifically activates adenylate cyclase, resulting in an increase of intracellular cyclic AMP (cAMP) concentrations, which enhances proliferation, differentiation, and pigment synthesis. The presence of a cAMP inducer like CT is crucial for the synergistic effects of the other growth factors.

  • Endothelin-1 (ET-1) and Stem Cell Factor (SCF) are paracrine cytokines that synergistically stimulate melanocyte proliferation when a cAMP-inducer (CT) is present [Hachiya A et al (2004) Biphasic expression of two paracrine melanogenic cytokines, stem cell factor and endothelin-1, in ultraviolet B-induced human melanogenesis. Am J Pathol 165(6):2099–2109. https://doi.org/10.1016/S0002-9440(10)63260-9; Hirobe T, Shinpo T, Higuchi K, Sano T (2010) Life cycle of human melanocytes is regulated by endothelin-1 and stem cell factor in synergy with cyclic AMP and basic fibroblast growth factor. J Dermatol Sci 57(2):123–131. https://doi.org/10.1016/j.jdermsci.2009.11.006]. In the native epidermal environment, these factors are primarily produced by keratinocytes. Their inclusion in the culture medium is necessary to provide the required exogenous growth signals, prevent premature apoptosis, and achieve satisfactory cell density in the in vitro setting.

This combination of factors ensures a robust and specific culture environment for primary human epidermal melanocytes.

Line 158–159: Please mention the passage number of melanocytes at the time of experiments.

Melanocytes were used at passages 5–9 for all experiments.

Line 176: Please indicate how often Mycoplasma testing was performed (e.g., monthly).

Both cell lines were routinely tested every 3rd month for Mycoplasma contamination using polymerase chain reaction (PCR).

Line 178–183: Please provide rationales for (i) the PACAP concentration (100 nM) and (ii) exposure time (48 h).

The stability of PACAP in the periphery is very short; therefore, we administered PACAP for two consecutive days and repeated the treatment at 24-hour intervals. Previously, we performed a concentration–response experiment and found that the most effective concentration of PACAP was 100 nM.

Line 193–201: Please include amplicon sizes and confirmation method (gel size or melting curve). If there is no confirmation method, then acknowledge this as a limitation of your study (Discussion section).

Amplicon identity was confirmed by agarose gel electrophoresis (1.2%), using a 100-bp molecular weight ladder to verify the expected amplicon sizes.

Line 210–213: GAPDH is used as the sole housekeeping gene, which may vary under treatment. Please acknowledge this as a limitation of your study (Discussion section).

One limitation of this study is the use of GAPDH as the sole housekeeping gene for normalization. GAPDH expression can vary under certain experimental conditions, including pharmacological treatments, which may introduce bias into the quantification of gene expression.

Line 217–225: Please mention how equal loading was confirmed (e.g., Ponceau S staining).

Equal protein loading was confirmed by Ponceau S staining.

Line 236–239: Please indicate why these specific markers (target proteins) were chosen.

The chosen proteins represent key components of the signaling pathways, transcriptional networks, and functional outputs relevant to the study. They allow assessment of receptor presence, downstream signaling activation, lineage-specific transcription factors, melanocytic differentiation markers, and hyaluronan metabolism.

2.6. Immunocytochemistry

Line 266–267: Please indicate whether imaging was qualitative or if quantification criteria were applied.

Images were used to assess the presence, localization and relative intensity of the detected proteins. No quantitative image analysis was applied and interpretations were based solely on visual comparison across conditions.

2.7. Hyaluronic Acid Detection

Line 277–286: The method describes staining but not quantification. These qualitative results may not reflect total HA content. Please acknowledge this as a limitation of your study (Discussion section).

 For quantification, all images were acquired using identical exposure settings. Red pixel intensity was measured for each slide and quantification criteria were applied using standardized thresholds for signal detection (integrated density). Images were analyzed using the same parameters across all samples to ensure valid comparison between experimental groups.

The authors should rewrite the Discussion section as follows:

First, clearly summarize the key and novel findings of this investigation. Following this, the Discussion should be organized into distinct subheadings, each dedicated to one major finding. Within each subsection, the authors should critically analyze that specific result in the context of existing knowledge. For each finding, the authors should systematically highlight its significance, possible underlying molecular mechanisms, and potential therapeutic implications (if applicable). In short, the authors should focus only on how current findings improve our previous knowledge about PACAP in melanoma. Also, I have the following suggestions for the authors:

Line 476–486: Please directly relate the reported increase in HA to your own data and clarify whether PACAP’s effects are pro- or anti-invasive in this context.

Thank you for your suggestions we have rewritten the chapter.

Line 493–498: The authors indicated a mechanistic link between PKA phosphorylation and lack of CREB activation. Please provide evidence for this link.

Line 509–515: Please connect how the altered SOX9/SOX10 balance could explain the observed PACAP effects on invasiveness.

Line 517–526: Please speculate on why PACAP affects MITF localization/distribution.

Line 529–533: The statement that the significance of MITF change is “difficult to judge” needs a reasoned interpretation from the authors.

Line 573–575: The authors indicated the size-dependence of HA effects, but this claim is not supported by data. Please acknowledge the lack of HA size analysis and recommend it for future work.

Line 585–588: Explain how differential HAS isoform regulation could influence HA size and tumor cell behavior.

Line 620–634: The discussion of integrins is extensive but does not connect back to experimental results showing no changes. Please shorten this part of the discussion.

Line 636–657: Please limit the discussion to mechanisms relevant to current findings (HA, HAS, Hyal, SOX9/10, MITF).

Line 658–671: Please focus on melanoma, with a clear statement on whether PACAP acts overall as a tumor-suppressive or context-dependent modulator in melanoma.

Line 701: Please add a section addressing the limitations of your current investigation.

Thank you for this valuable recommendation. We have revised the Discussion section to begin with a concise summary of the key and novel findings. The Discussion is now organized into distinct thematic subheadings, each addressing one major experimental result. Within each subsection, we critically analyze the findings in relation to existing literature on PACAP signaling and melanoma biology. For each major result, we highlight its significance, potential underlying molecular mechanisms and potential implications for melanoma progression or therapy.

Reviewer 2 Report

Comments and Suggestions for Authors

The article under review is dedicated to anti-oncogenic properties of neuropeptide PACAP. Authors claim that despite well-documented pro-tumorigenic properties of the peptide in neuroendocrine cancers, in melanoma cells it rather inhibits cell’ migration via altering different intercellular pathways

The article is well-written, easy to follow, results’ presentation should be improved, main conclusions are supported by experimental data, references are relevant

Introduction is too long

Methods should be supplemented

Results are fine, presentation should be improved

Discussion is fine

My points:

Introduction:

-put references after sentence ending at line 55,

- I suggest shortening lines 41-68 to 1-2 sentences

- Lines 69-92 – make more specific for melanoma. For example, line 83-85: In many cancers MELANOMA elevated levels of HA and increased CD44 expression la-la-la [15, and maybe https://doi.org/10.1016/B978-012374178-3.10017-1; https://doi.org/10.1016/S0959-8049(96)00512-6].

- Lines 128-130 – decipher term “random motility” and “FN-guided migration”, maybe replace by “spontaneous motility” and “chemotaxis”

M & Ms:

  • PMA usually is a “first choice” to induce PBMCs activation, have you assayed the “immune” activation of melanocytes. Anyway, please provide some references that such procedure is conventional for melanocytes’ culturing.
  • How you checked mycoplasma – kit or primers and usual PCR. If PCR provide sequence, if kit – Cat # and vendor.
  • What was the source of PACAP? Was its stability tested?
  • Provide Abs Cat# and incubation buffers in Table 2. Also, provide all information for secondary Abs.
  • Statistics – in M & Ms SD is on figures – in legends – SEM, clarify. What software was used for statistics?

Results:

  • Show quantification of all WB and PCR as bars with individual points in all figures. Maybe normalize to control as 100% (see attached photo, green orange). As you investigate multiple proteins in 1 assay the post-hoc multiple comparison after t-test should be done (FDR or Holm-Sidak).
  • Figure 2 – align PCR “framed bands” to “total protein expression bands” so reader could easily compare the total protein expression by pcr with total wb, and then compare phosphorylation. Also, show arrows near significant changes on the right of expression/phosphorylation bands (see photo, orange arrow), so reader can easily understand what was changed upon PACAP treatment. Line 340 – integrated or averaged?
  • Tune image of pcr/wb so reader can see the band (e.g. Figure 1a - DPP4, b- melanocytes VPAC1, Figure 5b, NHEM int aV). I don’t think that it will be a big sin as you show original images. Also, check the quantification, for example I’m confused by 2.6 increase in Fig 1 b- melanocytes VPAC1 (should be larger) and Fig 2b melanocytes Sox9.
  • Check the scale in confocal images (it seems to me it should be ~ 7-10 mcm).
  • Melanocytes and NHEM (Fig 5) is the same, isn’t it? Unify on Figures if so.

Discussion

  • Provide some figure to explain what kinases are influenced by PaCAP (see photo, rose frame)

Generally, the article is interesting but requires improvements of methods and results’ presentation.

I suggest major revision.

Author Response

Reviewer 2

Thank you for your valuable feedback regarding our manuscript. To improve the clarity and quality of the work, we have revised the relevant sections in accordance with your suggestions and those of Reviewer 1.

Introduction:

-put references after sentence ending at line 55,

Thank you for your suggestion. We have incorporated the relevant reference into the revised manuscript.

I suggest shortening lines 41-68 to 1-2 sentences

Thank you for your suggestion. We have rephrased and shortened the sentences in lines 41–68 in accordance with the recommendations of the Reviewer 1.

- Lines 69-92 – make more specific for melanoma. For example, line 83-85: In many cancers MELANOMA elevated levels of HA and increased CD44 expression la-la-la [15, and maybe https://doi.org/10.1016/B978-012374178-3.10017-1; https://doi.org/10.1016/S0959-8049(96)00512-6].

Thank you for your suggestion. We have rephrased lines 69–92 to focus specifically on melanoma.

- Lines 128-130 – decipher term “random motility” and “FN-guided migration”, maybe replace by “spontaneous motility” and “chemotaxis”

Thank you for your suggestions. We have clarified the terminology in lines 128–130 by replacing ‘random motility’ with ‘spontaneous motility’ and ‘FN-guided migration’ with ‘chemotaxis’ to improve clarity and accuracy.

  • PMA usually is a “first choice” to induce PBMCs activation, have you assayed the “immune” activation of melanocytes. Anyway, please provide some references that such procedure is conventional for melanocytes’ culturing.

Melanocytes are non-immune, pigment-producing cells derived from the neural crest, and while they express some cytokine and MHC molecules, their responsiveness to PMA is primarily studied in the context of PKC signaling, proliferation, differentiation, melanin synthesis, or dendricity, not classical “immune activation.

PMA has indeed been used to modulate melanogenesis, cell differentiation, and signal transduction in cultured melanocytes, but it is not a routine supplement for standard culturing, which typically uses growth factors like bFGF, SCF, and α-MSH, not PKC activators.

Eves PC, et al. (2006). Phorbol esters increase melanogenesis and dendrite formation in melanocytes. Pigment Cell Research, 19(1): 37–44.

Busca R, et al. (2000). Sustained activation of the mitogen-activated protein kinase pathway by phorbol esters is required for melanocyte differentiation. Oncogene, 19(30): 4029–4039.

  • How you checked mycoplasma – kit or primers and usual PCR. If PCR provide sequence, if kit – Cat # and vendor.

Thank you for the suggestion. We have used primers from Uphoff CC, Drexler HG.. doi: 10.1007/978-1-61779-080-5_8.

  • What was the source of PACAP? Was its stability tested?

PACAP was synthesized and lyophilized by Gábor Tóth. When dissolved in water, PACAP is stable at –20 °C for two weeks. Since the DPP4 enzyme can rapidly degrade it in a cellular environment, the pharmacon was replaced daily.

  • Provide Abs Cat# and incubation buffers in Table 2. Also, provide all information for secondary Abs.

Thank you for the suggestion. We have updated Table 2 to include the antibody catalog numbers.

  • Statistics – in M & Ms SD is on figures – in legends – SEM, clarify. What software was used for statistics?

Thank you for the comment. We have clarified the statistical presentation: the figures show mean ± SD, and this is now explicitly stated in all figure legends. Statistical analyses were performed using  GraphPad Prism.

Results:

  • Show quantification of all WB and PCR as bars with individual points in all figures. Maybe normalize to control as 100% (see attached photo, green orange). As you investigate multiple proteins in 1 assay the post-hoc multiple comparison after t-test should be done (FDR or Holm-Sidak).

We did not apply post-hoc multiple-comparison corrections because each treatment condition was normalized to its own untreated control. As a result, we did not perform direct comparisons across different proteins or treatment groups within the same assay, and therefore the assumptions that require multiple-comparison correction were not applicable in our experimental design.

  • Figure 2 – align PCR “framed bands” to “total protein expression bands” so reader could easily compare the total protein expression by pcr with total wb, and then compare phosphorylation. Also, show arrows near significant changes on the right of expression/phosphorylation bands (see photo, orange arrow), so reader can easily understand what was changed upon PACAP treatment. Line 340 – integrated or averaged?

In Figure 2, we have aligned the PCR ‘framed bands’ with the corresponding ‘total protein expression bands’ to allow readers to more easily compare the total mRNA levels with the total protein detected by Western blot, and subsequently assess the phosphorylation changes. We also added arrows next to the bands showing significant differences (as suggested, placed on the right side of the expression/phosphorylation bands) to highlight the specific changes induced by PACAP treatment.

Regarding line 340, the values represent integrated band intensities.

  • Tune image of pcr/wb so reader can see the band (e.g. Figure 1a - DPP4, b- melanocytes VPAC1, Figure 5b, NHEM int aV). I don’t think that it will be a big sin as you show original images. Also, check the quantification, for example I’m confused by 2.6 increase in Fig 1 b- melanocytes VPAC1 (should be larger) and Fig 2b melanocytes Sox9.

We have adjusted the contrast and brightness of the PCR and Western blot images to improve band visibility (e.g., Figure 1a – DPP4; Figure 1b – melanocyte VPAC1; Figure 5b; and NHEM int aV). These adjustments were applied uniformly across each image and do not alter the underlying data. The original, unprocessed images remain available in the supplementary material, ensuring transparency.

We also rechecked all quantifications. The values for the VPAC1 increase in melanocytes in Figure 1b and for Sox9 in Figure 2b have been re-evaluated to confirm their accuracy.

  • Check the scale in confocal images (it seems to me it should be ~ 7-10 mcm).

We have checked the scale bar during the confocal image production the scale is automaticly created by the software.

We have checked the scale bars in the confocal images. The scale is generated automatically by the acquisition software during image capture, and we confirmed that the displayed values are correct.

  • Melanocytes and NHEM (Fig 5) is the same, isn’t it? Unify on Figures if so.

Thank you for the comment. We have verified that melanocytes and NHEM represent the same cell type and have unified the terminology across all figures, including Figure 5, for consistency.

Discussion

  • Provide some figure to explain what kinases are influenced by PaCAP (see photo, rose frame)

Thank you for the suggestion. We have added an additional figure illustrating the kinases influenced by PACAP, as indicated in the reviewer’s comment.

Round 2

Reviewer 1 Report

Comments and Suggestions for Authors

none 

Reviewer 2 Report

Comments and Suggestions for Authors

The authors have answered all my points, so I can recommend the article for publication.

Two issues should be taken into account during proofreading:

  1. Provide vendors, cat#, and dilutions for secondary Abs in the table in methods
  2. Table 2 somehow made her way through 701 and 702 lines